# Modification of a Conventional Deep Learning Model to Classify Simulated Breathing Patterns: A Step toward Real-Time Monitoring of Patients with Respiratory Infectious Diseases

**DOI:** 10.3390/s23125592

**Published:** 2023-06-15

**Authors:** Jinho Park, Aaron James Mah, Thien Nguyen, Soongho Park, Leili Ghazi Zadeh, Babak Shadgan, Amir H. Gandjbakhche

**Affiliations:** 1Eunice Kennedy Shriver National Institute of Child Health and Human Development, National Institutes of Health, 49 Convent Dr., Bethesda, MD 20894, USA; jinho.park@nih.gov (J.P.); thien.nguyen4@nih.gov (T.N.); soongho.park@nih.gov (S.P.); 2Implantable Biosensing Laboratory, International Collaboration on Repair Discoveries, Vancouver, BC V5Z 1M9, Canada; aamah@student.ubc.ca (A.J.M.); lili.ghazi@gmail.com (L.G.Z.); babak.shadgan@ubc.ca (B.S.); 3Department of Pathology & Laboratory Medicine, University of British Columbia, Vancouver, BC V6T 1Z7, Canada

**Keywords:** COVID-19, deep learning, convolutional neural network, respiratory disease, NIRS, wearable device

## Abstract

The emergence of the global coronavirus pandemic in 2019 (COVID-19 disease) created a need for remote methods to detect and continuously monitor patients with infectious respiratory diseases. Many different devices, including thermometers, pulse oximeters, smartwatches, and rings, were proposed to monitor the symptoms of infected individuals at home. However, these consumer-grade devices are typically not capable of automated monitoring during both day and night. This study aims to develop a method to classify and monitor breathing patterns in real-time using tissue hemodynamic responses and a deep convolutional neural network (CNN)-based classification algorithm. Tissue hemodynamic responses at the sternal manubrium were collected in 21 healthy volunteers using a wearable near-infrared spectroscopy (NIRS) device during three different breathing conditions. We developed a deep CNN-based classification algorithm to classify and monitor breathing patterns in real time. The classification method was designed by improving and modifying the pre-activation residual network (Pre-ResNet) previously developed to classify two-dimensional (2D) images. Three different one-dimensional CNN (1D-CNN) classification models based on Pre-ResNet were developed. By using these models, we were able to obtain an average classification accuracy of 88.79% (without Stage 1 (data size reducing convolutional layer)), 90.58% (with 1 × 3 Stage 1), and 91.77% (with 1 × 5 Stage 1).

## 1. Introduction

Infectious respiratory diseases are caused by various microorganisms such as viruses, fungi, parasites, and bacteria [1,2]. Examples of infectious respiratory diseases include tuberculosis, diphtheria, bacterial pneumonia, and viral pneumonia, such as influenza and COVID-19 disease [3,4,5]. These diseases affect not only individuals but can also have a significant impact on society, depending on the extent. COVID-19 disease was declared a global pandemic in 2020 by the World Health Organization (WHO) and has had a great economic and social impact worldwide [5]. To prevent the severe consequences of infectious disease, it is of utmost importance to develop an effective method to monitor infected individuals. The current practice of treating individuals with infectious respiratory disease requires in-person examination, chest radiography, and, if necessary, blood and sputum tests [6,7,8]. In a global pandemic, it is impossible to monitor individual patients due to the large number of patients and limited medical personnel [9,10]. Additionally, in the case of highly contagious infectious diseases, in-person examination poses an increased risk of infection to healthcare providers [11]. Therefore, there is a need for remote methods to monitor patients with infectious diseases to facilitate more efficient treatments and prevent the spread of infection [12,13].

Common features of infectious respiratory diseases are symptoms caused by abnormalities in the respiratory system, such as cough and rapid and shallow breathing (tachypnoea) [14,15]. These symptoms are often accompanied by fever, headache, fatigue, and lethargy [16,17]. To examine these symptoms without direct contact with the patient, different methods are being conducted to analyze abnormal breathing patterns. Among these methods, radar [18,19], CT scans [20], X-rays [21], and ultrasounds [22] have shown promising results. However, limitations regarding accessibility, patient movement, and high costs prevent these techniques from being suitable for real-time and continuous patient monitoring [18,19,20,21,22,23,24,25]. To overcome these difficulties, various wearable biosensors that can test and monitor patients in real-time are being developed [26]. Among them, near-infrared spectroscopy (NIRS) is receiving great attention due to its simple optical device structure, low-cost, and capability to non-invasively monitor changes in tissue hemodynamic and oxygenation responses to respiratory infections [27]. In addition to a conventional provision of heart rate and respiratory rate, commercially available wearable sensors such as smart watches can measure peripheral arterial blood oxygen saturation (SpO_2_), which was reported as a critical indicator of deterioration in patients with infectious respiratory diseases [28]. The commercial wearable smart sensor can provide convenience when observing biological signals, but it can cause false readings due to various factors including poor circulation, skin thickness, and skin color [29]. While SpO_2_ is measured from arterial blood circulation, NIRS can measure changes in hemoglobin concentration within tissue microvasculature. This provides a significant advantage to standard pulse oximeters, as venous capillary blood provides the majority of the contribution to the hemoglobin absorption spectrum [30]. Furthermore, NIRS can measure tissue oxygenation without pulsatile flow, which provides a significant advantage in the critically ill patient monitoring [31]. Additionally, Cheung et al. found that transcutaneous muscle NIRS can detect the effects of hypoxia significantly sooner than pulse oximetry [32]. In patients with severe respiratory disease, early diagnosis and treatment are essential to ensure improved patient prognosis and reduce long-term negative health consequences [33]. Hence, in order to detect the effects of infectious respiratory diseases on patients earlier, a NIRS device was used to measure tissue hemodynamics.

In our previous studies, we employed a wearable NIRS device to monitor tissue hemodynamic responses, including changes in tissue oxygenated hemoglobin (O_2_Hb), deoxygenated hemoglobin (HHb), total hemoglobin (THb), and tissue saturation index (TSI) from the chest of healthy volunteers during different simulated breathing tasks [34]. Measured O_2_Hb signals were then processed to extract three features: O_2_Hb change, breathing interval, and breathing depth, averaged over a period of 60 s. These features were then fed to a well-known machine learning model (Random Forest classification) to classify different simulated breathing tasks: baseline (rest), rapid/shallow, and loaded breathing. We were able to achieve a classification accuracy of 87%. However, a drawback that can hinder the real-time monitoring capability of this methodology is the extra step of feature extraction. In this study, to eliminate this time-consuming step, we propose using a deep learning model to classify the three breathing patterns.

When the AlexNet model based on deep CNN won the 2012 ImageNet Large Scale Visual Recognition Challenge (ISLVRC) with an overwhelming performance [35], various types of convolutional neural network (CNN) models were proposed in various fields, including image classification, image enhancement, computer vision, medical imaging, and network security. The development of these deep learning algorithms has high potential for applications in the classification and monitoring of infectious respiratory diseases. Cho et al. used a thermal camera to track breathing patterns using temperature changes around an individual’s nose [36]. Following this study, a CNN-based algorithm was used to classify psychological stress levels automatically. Shah et al. proposed a method to identify characteristic patterns of COVID-19 disease from CT scan images taken using a deep learning algorithm [37]. Chen proposed a deep learning model that adds a 3D attention module to the 3D U-Net model [38], enabling the segmentation of COVID-19 lung lesions from CT images [39]. Rabbah et al., Haritha et al., and Qjidaa et al. proposed a method of detecting COVID-19-infected individuals from chest X-ray images using a CNN-based algorithm [40,41,42]. Wang et al., used a depth camera to measure depth variations in the chest, abdomen, and shoulder of participants to classify six breathing patterns (Eupnea, Tachypnea, Bradypnea, Biots, Cheyne-Stokes, and Central-Apnea) using the BI-AT-GRU algorithm, which combines bidirectional and attentional mechanisms in a Gated Recurrent Unit neural network [15]. Sarno et al. proposed a method that utilizes an electronic nose (E-nose) to collect sweat samples from the human axilla and employed a stacked Deep Neural Network to classify individuals as having or not having respiratory conditions [43].

All of these methods showed promising results for classifying patients suffering from respiratory disease. However, the data acquisition method is unsuitable for continuous monitoring due to the constant inconvenience to the patients. To solve this problem, we aimed to apply a deep learning algorithm to the data acquired from a wearable NIRS device. We hypothesize that the capability of a deep learning model to enact automatic feature extraction will enable the use of a wearable NIRS device for real-time monitoring and increase classification accuracy.

## 2. Materials and Methods

### 2.1. Data Collection

Tissue hemodynamic indices, including O_2_Hb, HHb, THb, and TSI were measured at the manubrium using a wearable NIRS device. The NIRS device consists of three light sources, emitting light at two wavelengths (760 nm and 850 nm), and one light detector. Data were collected from 21 healthy adult volunteers (12 males and 9 females; mean age = 29.47 ± 9.73 years old) through a clinical protocol approved by the Clinical Research Ethics Board at the University of British Columbia. Each participant performed three separate breathing conditions: baseline (3 min), loaded (5 min), and rapid/shallow (5 min). During the baseline phase, participants breathed normally through the nose or mouth at a relaxed pace of breathing. To simulate dyspnea, a common breathing issue observed during acute pneumonia, a respiratory trainer was used during the loaded breathing phase. The respiratory trainer increases resistance during breathing, forcing respiratory muscles to work harder to facilitate breathing. The rapid/shallow breathing phase was designed to simulate tachypnea—the condition of being unable to take deep breaths during acute pneumonia. Participants were instructed to breathe 25 times per minute during this phase. The data acquisition rate was 10 Hz. The signals acquired through a NIRS device are obtained in the form of a one-dimensional signal. Figure 1 shows the O_2_HB signal data obtained using the NIRS device during the three phases. During the three phases, the signal obtained under the loaded phase has the largest amplitude, while the signal obtained under the rapid phase has the shortest period. The data for each condition is cropped at 6.4-s intervals and used as input data for model training and testing. Further description of data collection can be found in our previous publication [34].

### 2.2. Classification Model for Simulated Breathing Model

In this paper, we propose a classification model based on deep CNN to classify simulated breathing patterns. The classification algorithm was developed by improving and modifying the pre-activation residual model (Pre-ResNet)—a two-dimensional (2D) image classification algorithm [44]. The parameters of CNN are learned by stochastic gradient descent (SGD) [45] with a backpropagation [46]. However, this approach presents a vanishing gradient problem—where the gradient becomes smaller as the depth of the network increases [47,48,49]. To solve this problem, He et al. proposed a residual unit that adds a shortcut connection between building blocks for the residual learning [44]. The building block is proposed to increase the depth of the network and is composed of standardized layers. Short connections mitigate gradient loss by skipping one or more layers during backpropagation. This short connection has been applied to various CNN models [50,51,52]. The residual unit is defined as follows:(1)$$x_{l+1}=F\left(x_{l},W_{l}\right)+x_{l},$$
where $x_{l}$ and $x_{l+1}$ represent the input and output features of l-th units, respectively. $W_{l}$ is the weight parameters of the residual unit, and F is the residual function. The residual function includes a convolutional layer, an active function (ReLU) [53], and a batch normalization (BN) [54] as shown in Figure 2b.

The Pre-ResNet model is designed for a 2D signal with a size of 32 × 32. To apply this model to the 1D signal obtained from the NIRS device, the 2D convolutional layer was changed to a 1D convolutional layer. Additionally, we added a residual unit consisting of a 1 × 5 kernel to create a global feature while reducing the data size at the front of the network. We confirmed that adding a residual unit with a 1 × 5 kernel shows better classification performance than simply downsampling the data through experiments. Figure 2a shows the architecture of the CNN model for the classification of breathing patterns. The detailed parameter information is shown in Table 1. BN and ReLU functions are omitted from the table. Each convolution block belonging to the same group has the same kernel size and number of kernels. The block structure represents the kernel size and number of the convolutional layer included in each block. Block number indicates the number of residual units constituting each group. The input size and output size indicate the size of the input feature and output feature of the stage, respectively.

As the depth of the network increases, downsampling by 1/2 was performed using a stride of 2, and the feature map dimension is doubled in the first convolutional layer of the first residual unit in Stage 1, Stage 3, and Stage 4 to reduce the feature map and generate high-dimensional feature maps. The parameter optimization of the network was performed using SGD. SGD is an iterative optimization algorithm widely used in machine learning applications to find model parameters that minimize the error between predicted output and the ground truth value. The gradient descent algorithm computes the gradient using the entire training dataset for each iteration to update the model. In contrast, SGD uses a single random training example to compute the gradient. As a result, SGD exhibits faster learning speed compared to gradient descent. Cross entropy loss was used as the loss function.

## 3. Results

We used data obtained from a customized wearable NIRS device for a classification experiment. The outputs of the NIRS device are O_2_Hb, HHb, THb, and TSI. A customized data acquisition software collects and converts measured signals to changes in these outputs [34]. Collected data were preprocessed, and signals containing movement artifacts were removed. The preprocessed signal is cropped at 64 data intervals, which is equivalent to a signal length of 6.4 s. As a result, 531 baseline samples, 780 loaded samples, and 874 rapid/shallow samples were generated. The entire dataset was randomly selected at a ratio of 80:20 for model training and evaluation. For classification, only O_2_Hb and HHb datasets were used. The classification did not include THb (total hemoglobin, which is the sum of O_2_Hb and HHb) and TSI (the ratio of O_2_Hb to THb) because they are directly affected by changes in O_2_Hb and HHb. Table 2 displays the number of O_2_Hb and HHb datasets used for training and testing.

The classification model used in this experiment consists of a total of 113 layers, as shown in Table 1. Our classification model was trained using SGD with a batch size of 64 samples and momentum of 0.9 for 120 epochs. The initial learning rate started at 0.1 and was divided by a factor of 10 every 30 epochs. All classification models used in the experiment were trained from scratch.

Table 3 shows the test results of the classification model. Each CNN-based method was tested five times, except for the Random Forest method, and Table 3 shows the average accuracy, standard deviation (STD), and best accuracy. The accuracy of the results can be calculated using Equation (2).
(2)$$\mathrm{Accuracy}(\%)=\frac{\mathrm{Total}\;\mathrm{number}\;\mathrm{of}\;\mathrm{true}\;\mathrm{positive}}{\mathrm{Total}\;\mathrm{number}\;\mathrm{of}\;\mathrm{dataset}}\times 100\%,$$

The Random Forest method is a classification model we have tested in previous research [34]. This model contains 100 trees, and the characteristics of the average amplitude, interval, and magnitude of the signal are used for training and evaluation of the model. The Pre-ResNet with DS shows the test result after downsampling the signal to 1/2 size without Stage 1.

As a result of the experiment, the CNN-based classification model shows better overall performance than the Random Forest method. Adding Stage 1 improves the classification performance compared to direct downsampling of the signal. Additionally, the CNN-based model using a 1 × 5 kernel in Stage 1 achieved the highest performance on the O_2_Hb dataset, with an accuracy of 92.43%. In the experiments using Pre-ResNet with downsampling and Pre-ResNet with Stage 1 (1 × 5 kernel), classification experiments were compared using individual signals of O_2_Hb and HHb, as well as combined signals of HHb and O_2_Hb, and it was found that the best performance was achieved when using only the O_2_Hb data.

Figure 3 shows the normalized confusion matrix of different CNN-based methods. The true label represents the label of the data sample, and the predicted label represents the label estimated from the model. It can be seen that the Loaded class has the lowest classification accuracy across all three different models.

Table 4 shows the recall values of each class and the balanced accuracy of best classification accuracies after five-times testing. The balanced accuracies in Table 4 show a similar performance to the best accuracies in Table 3.

Table 5 shows the results of the classification experiment conducted using the dataset split on a participant basis. Out of 21 participant datasets, 17 participant datasets were randomly selected for model training, and the remaining 4 participant datasets were used for classification testing. The classification of the datasets split on the subject level yields slightly less accurate results compared to the classification of the datasets of all the subjects.

To compare the classification performance of our proposed method with other advanced deep learning classification algorithms, we used EfficientNetV2 m [55], PyramidNet [51], and CF-CNN [52]. EfficientNetV2 m and PyramidNet models are CNN-based algorithms developed for image classification and are being utilized in various fields [56,57,58,59,60].

CF-CNN proposed a classification model with multiple coarse sub-networks and a multilevel label augmentation method to enhance the training performance of the base model. For comparison experiments, we replaced the 2D convolutional layer of this model with a 1D convolutional layer. PyramidNet is composed of 272 layers, and the widening factor is set to 200. To train and test the CF-CNN model, we used PyramidNet as the base model. The layers of coarse 1, coarse 2, and the fine network were set to 26, 52, and 272, respectively, with group labels of 1, 2, and 3.

Each classification model was tested five times and trained from scratch. The model training parameters and input data size were set as used in the experiment that yielded the results in Table 1 and used for training. The experimental results show that, despite having a smaller number of parameters and floating point operations per second (FLOPS), the Pre-ResNet model with Stage 1 (1 × 5) demonstrates similar classification performance to the EfficientNetV2 m model.

## 4. Discussions

The authors of this study modified a conventional deep convolutional neural network (CNN) to classify three different breathing patterns: baseline, loaded, and rapid/shallow breathing. Since the CNN shares parameters, they can create robust features for shifted data. In addition, the CNN performs convolution operation using the kernel; hence, it shows good performance in detecting repetitive patterns. As a result, we were able to obtain a high classification accuracy when using the CNN (92.43%) in Table 1.

In this experiment, data sampled from all participants were randomly divided into training and testing sets at an 80:20 ratio for classification experiments. The experimental results showed that the classification models based on 1D-CNN (92.43%) yielded much better performance than the classification model using Random Forest (87%). To obtain good classification performance of traditional machine learning methods such as Random Forest, features and classification methods that can distinguish breathing patterns are required. However, it is not easy to generate various features and/or classification models. The CNN-based classification model demonstrates increased performance because it automatically generates numerous features and performs classification simultaneously during the learning process. Additionally, when Stage 1 using the 1 × 5 kernel was added, it showed better classification performance than Stage 1 using the 1 × 3 kernel due to its ability to detect important features in a wider temporal domain. In Figure 3, the data of the loaded class showed the lowest classification accuracy due to the intermediate characteristics between the baseline and rapid/shallow respiration.

The results of Table 3 reveal that solely employing the O_2_Hb data type yields higher accuracy in comparison to utilizing both O_2_Hb and HHb data. This can be related to the characteristics of the data. The O_2_Hb signal has a higher signal-to-noise ratio and acceptable high reproducibility compared to the HHb signal [61,62,63]. Additionally, lower classification accuracy when using both O_2_Hb and HHb could be because HHb contains additional information other than respiration. O_2_Hb is formed when hemoglobin combines with oxygen molecules during the process of respiration. On the other hand, HHb is a protein that releases the oxygen molecules it was carrying and travels back to the lungs to pick up more oxygen [64]. HHb is indirectly linked to oxygenation and can be influenced by factors such as changes in blood flow, vascular conditions, and tissue metabolism, which are not solely related to respiration.

Table 4 demonstrates that the similarity between the general accuracy and the balanced accuracy confirms that the classification model is well-trained regardless of the difference in the number of data in each class. Based on our results, it can be confirmed that the O_2_Hb data obtained using the NIRS device has suitable characteristics for classifying breathing patterns. Furthermore, by automatically generating appropriate features for classification, the CNN-based algorithm shows better performance than the Random Forest algorithm, which uses handcraft-based features (average amplitude, interval, and magnitude).

The high classification accuracies obtained when datasets were divided at the participant level (Table 5) prove the generalizability of our proposed model. Additionally, the performance of our proposed model was compared with the following state-of-the-art models: EfficientNetV2-M, PyramidNet, and CF-CNN. From the experimental results, it was confirmed that the proposed method shows similar performance to the EfficientNetV2 m model while using much fewer parameters and FLOPS compared to other models.

While this study has yielded promising results, it is important to acknowledge a notable limitation. Specifically, the NIRS respiratory data utilized in our research were obtained exclusively from healthy individuals. Consequently, the simulated symptoms of acute respiratory diseases such as COVID-19 and viral pneumonia were based on loaded breathing and rapid/shallow breathing patterns. To enhance the classification method, future studies should aim to broaden the scope by collecting respiratory signals from individuals diagnosed with acute pneumonia and incorporating this data into the analysis. This would provide valuable insights and improve the applicability of our findings.

## 5. Conclusions

In this study, we proposed a CNN-based method to classify respiratory patterns in patients with infectious respiratory diseases. This method employs oxygenated hemoglobin change measured with a wearable NIRS device as the input. The wearable NIRS device used for data acquisition is small, portable, and attachable to the human body. Additionally, the NIRS device has no environmental constraints, which allows for continuous monitoring. We designed a 1D-CNN-based classifier by improving and modifying the pre-activation residual network developed for 2D image classification to classify respiratory patterns. With the developed classification model, we were able to obtain a maximum classification accuracy of 92.43%. The proposed method can be used for the remote detection and real-time monitoring of various respiratory diseases, including COVID-19, tuberculosis, influenza, and pneumonia.

## Figures and Tables

**Figure 1 sensors-23-05592-f001:**
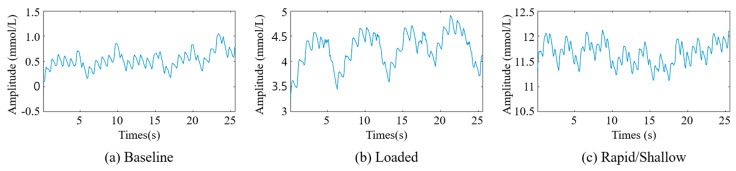
O_2_Hb signal during three breathing phases.

**Figure 2 sensors-23-05592-f002:**
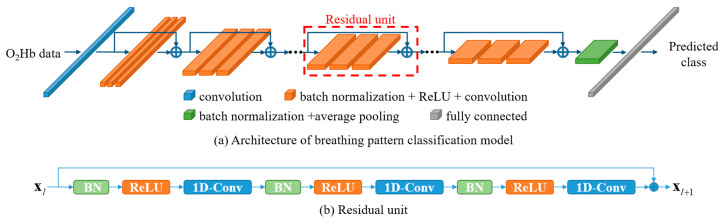
Breathing pattern classification model. (**a**) The framework of the proposed method: the proposed CNN model consists of multiple residual units. (**b**) The structure of a residual unit.

**Figure 3 sensors-23-05592-f003:**

Normalized confusion matrix for the CNN based models on the test set.

**Table 1 sensors-23-05592-t001:** Classification model architectures of a CNN with 1 × 5 kernel Stage 1.

Group Name	Input Size Output Size	Block Structure (Kernel Size, Number)	Block Numbers (113-Layers)
Stage 0	1 × 64	1 × 5, 16	1
Stage 1	1 × 64 1 × 32	1 × 1, 16	1
		1 × 5, 16	
		1 × 1, 16	
Stage 2	1 × 32 1 × 32	1 × 1, 16	12
		1 × 3, 16	
		1 × 1, 64	
Stage 3	1 × 32 1 × 16	1 × 1, 32	12
		1 × 3, 32	
		1 × 1, 128	
Stage 4	1 × 16 1 × 8	1 × 1, 64	12
		1 × 3, 64	
		1 × 1, 256	
Average pooling	1 × 8 1 × 256	1 × 8	1
Fully connected layer	1 × 256 Classes number		1

**Table 2 sensors-23-05592-t002:** The number of the datasets.

Class	Train	Test
Baseline	425	106
Loaded	624	156
Rapid/shallow	700	174

**Table 3 sensors-23-05592-t003:** Classification results. Where DS denotes downsampling.

Method	Data Type	Mean Accuracy (%)	STD	Best Accuracy (%)
Random Forest	O_2_Hb	-	-	87.00
Pre-ResNet with DS	O_2_Hb	88.79	0.423	89.44
	HHb	86.28	0.550	87.16
	O_2_Hb and HHb	88.02	0.490	88.76
Pre-ResNet with Stage 1, (1 × 3)	O_2_Hb	90.58	0.488	91.51
	HHb	89.63	0.639	90.37
	O_2_Hb and HHb	90.23	0.658	91.28
Pre-ResNet with Stage 1, (1 × 5)	O_2_Hb	91.77	0.456	92.43
	HHb	89.68	0.490	90.60
	O_2_Hb and HHb	90.78	0.523	91.74

**Table 4 sensors-23-05592-t004:** Balanced accuracy of each CNN based methods on the O_2_Hb dataset.

Metric	Pre-ResNet with DS	Pre-ResNet with Stage 1, (1 × 3)	Pre-ResNet with Stage 1, (1 × 5)
Recall Baseline	0.92	0.94	0.93
Recall Loaded	0.81	0.85	0.90
Recall Rapid	0.96	0.95	0.94
Balanced Accuracy	89.66%	91.33%	92.33%

**Table 5 sensors-23-05592-t005:** Classification results on the dataset split on the subject level.

Method	Data Type	Number of Parameters	FLOPS	Mean Accuracy (%)	STD	Best Accuracy (%)
Pre-ResNet with DS	O_2_Hb	0.7 M	15 M	87.25	0.472	88.07
Pre-ResNet with Stage 1, (1 × 5)		0.7 M	15 M	90.14	0.649	91.28
EfficientNetV2 m with DS [55]		52 M	225 M	91.05	0.562	91.97
PyramidNet with DS [51]		17 M	396 M	87.39	0.481	88.89
CF-CNN with DS [52]		29.7 M	627 M	89.27	0.531	90.74

## Data Availability

Our code and data available at https://github.com/dkskzmffps/BPC_NIRSdata, Accessed on 7 June 2023.

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
