# Peer review of "Modification of a Conventional Deep Learning Model to Classify Simulated Breathing Patterns: A Step toward Real-Time Monitoring of Patients with Respiratory Infectious Diseases"

_sensors, 2023, doi:10.3390/s23125592_

Round 1
Reviewer 1 Report
The author proposes a CNN based method to classify the respiratory patterns of infectious disease patients. Realized remote detection and real-time monitoring of various respiratory system diseases. However, some additional clarification and experiment of the paper would be beneficial for the reader's understanding. They are outlined below:
1. In the introduction section, lines 42 to 43, the author emphasizes the development of an effective method for screening infected individuals in order to prevent the serious consequences of infectious respiratory diseases on society. As far as I know, take COVID-19 as an example (the author also mentioned the disease in the key words), there are more effective non-contact screening methods, such as antigen detection. I suggest that the author focus on low-cost remote real-time respiratory monitoring for patients with infectious respiratory diseases.
2. In the final paragraph of the introduction, the author introduces the enormous potential of deep learning in classifying and monitoring infectious respiratory diseases. I suggest adding some other applications of deep learning in respiratory diseases, such as segmentation. Here are some references. (IEEE Transactions on Industrial Informatics, 2021, 17(9): 6528-6538, An effective deep neural network for lung lesions segmentation from COVID-19 CT images; IEEE Transactions on Industrial Informatics, 2022, Detection of Infectious Respiratory Disease Through Sweat From Axillary Using an E-Nose With Stacked Deep Neural Network.)
3. I suggest that the expression of (c) in Figure 1 should be consistent with other parts of the text. Changing to (c) 'Rapid/Shallow' may be better.
4. I suggest providing some explanations for SGD on lines 173 and 187, rather than using abbreviations directly.
5. The author summarizes the classification experiments using O2Hb and HHb alone, as well as the combined signals of O2Hb and HHb, in Table 3. Surprisingly, the best results were achieved when using O2Hb data alone. I think the author can discuss this issue. Why does the performance actually decrease after adding information?
6. In Table 3, why did the ‘Pre-ResNet with Stage 1, (1×3)’ section not conduct experiments on individual HHb and the combination of O2Hb and HHb?
7. There are some errors in the text explanation section of Figure 3. It duplicates the explanation in Figure 2 and does not explain Figure 3.
8. I think in Eq. (2) for calculating accuracy, it should not be multiplied by 100. Because the unit used in expressing accuracy data in the text is '% '. That is to say, the accuracy formula should calculate a decimal number between 0 and 1.
9. In Table 5, the author presents the comparison effect between the designed model and the EfficientNetV2-M model. But I suggest adding more comparative experiments to fully demonstrate the superiority of the method.
10. The author emphasized the real-time nature of the method in the title and abstract sections of the article. This is necessary for respiratory monitoring of infectious respiratory diseases. However, in the experimental section of the article, the real-time performance of the method was not reflected except for the number of method parameters specified in Table 5. I suggest the author conduct supplementary experiments on this issue.
Moderate editing of English language required.
Author Response
Dear Reviewers,
We highly appreciate the valuable comments and suggestions, which contributed to improve the quality of our manuscript. We have carefully considered and revised the manuscript according to reviewer’s comments. We submit replies to reviewer comments. In this letter, the reviewers’ comments are in blue, followed by our response in black. We also provide the yellow-highlighted version of the revised manuscript highlighting our changes. And we checked the sentences of the manuscript again. We believe that we have answered questions, addressed comments, and applied all suggestions given by the reviewers.

Reviewer 2 Report
1) Are the three different breathing patterns (baseline, loaded, and rapid/shallow) sufficient to determine respiratory infections?
2) Is it possible to differentiate the different infections (COVID, Influenza, RSV, etc.) with the developed system?
3) The experiments were performed on 21 healthy volunteers. To prove that the system can be applied to detect respiratory infections, wouldn’t it be necessary to apply it to infected individuals?
Author Response

(The authors gave the same response as above.)

Round 2
Reviewer 1 Report
The authors have solved all my problems.